# A Comparison between Inbred and Hybrid Maize Haploid Inducers

**DOI:** 10.3390/plants12051095

**Published:** 2023-03-01

**Authors:** Henrique Uliana Trentin, Recep Yavuz, Abil Dermail, Ursula Karoline Frei, Somak Dutta, Thomas Lübberstedt

**Affiliations:** 1Bayer Crop Science, Coxilha 99145-000, RS, Brazil; 2Department of Agronomy, Iowa State University, Ames, IA 50011, USA; 3Department of Agronomy, Faculty of Agriculture, Khon Kaen University, Khon Kaen 40002, Thailand; 4Department of Statistics, Iowa State University, Ames, IA 50011, USA

**Keywords:** *Zea mays* L., hybrid inducer, heterosis, haploid induction rate, seed set, agronomic performance

## Abstract

The effectiveness of haploid induction systems is regarded not only for high haploid induction rate (HIR) but also resource savings. Isolation fields are proposed for hybrid induction. However, efficient haploid production depends on inducer traits such as high HIR, abundant pollen production, and tall plants. Seven hybrid inducers and their respective parents were evaluated over three years for HIR, seeds set in cross-pollinations, plant and ear height, tassel size, and tassel branching. Mid-parent heterosis was estimated to quantify how much inducer traits improve in hybrids in comparison to their parents. Heterosis benefits hybrid inducers for plant height, ear height, and tassel size. Two hybrid inducers, BH201/LH82-Ped126 and BH201/LH82-Ped128, are promising for haploid induction in isolation fields. Hybrid inducers offer convenience and resource-effectiveness for haploid induction by means of improving plant vigor without compromising HIR.

## 1. Introduction

Doubled haploid (DH) technology is based on in vivo maternal haploid induction and is widely applied in maize breeding [1]. Through in vivo maternal system, haploid induction requires haploid inducers, a male genotype with the ability to induce haploids, and source germplasm as donor genotypes. Most inducers known today are inbreds that breed true and are uniform, facilitating maintenance and management. Inbreds provide simplified logistics in comparison to hybrids by avoiding the concomitant maintenance of parental inbreds and hybrid seed production. However, inbreeding depression as well as a higher susceptibility to diseases [2] and limited seed sets [3] are prime flaws.

One way of overcoming this is by using hybrid inducers. In hybrid breeding, multiple traits important to inducers, such as tassel size, pollen production, and disease resistance can be improved. While this is common for improving maize grain yield due to heterosis, no hybrid vigor has been observed for HIR [4], consistent with the fact that this is a gametophytic trait [5,6,7]. A major challenge in hybrid inducers is that both parents must be fixed for the same marker traits; otherwise, the differentiation of haploid and diploid plants based on those characteristics becomes unreliable. If the discrimination of haploid and diploid seeds is based on oil content (OC), two genetically distinct parents with similarly high OC levels need to be developed to ensure accurate seed discrimination. Developing such parents would be both challenging and time-consuming. However, if discrimination is based on the *R1-nj* anthocyanin kernel marker, all inducer typescarrying this single gene are equally suitable. Another challenge associated with hybrid seed production is the development and maintenance of separate germplasm pools. Moreover, employing hybrid inducers requires the continuous production of both parent and hybrid seeds. As hybrids are taller than inbreds and synthetics, more attention must be paid to lodging tolerance [8]. 

Although most available inducers are inbred lines [9], there are reports on hybrids such as RWS/RWK-76 [10], TAILs [11], and 2GTAILs [12]. Hybrid inducers showed better agronomic performance in target environments, indicating that the exploitation of heterosis on agronomic traits including plant stature, flowering behaviors, and seed sets is feasible. To date, maternal haploid induction can be performed through either induction nurseries or isolation fields. Each of those methods requires distinct haploid inducer ideotypes. The former method is suitable for shorter inducers with breast-high tassel position and good tassel bending whereas the latter requires taller inducers with high pollen production [13]. The Doubled Haploid Facility at Iowa State University has released haploid inducer BHI201 possessing an average HIR of 12–14% [9], and several hybrid inducers were derived from that founder line. Therefore, this study aimed to (i) investigate the phenotypic differences between inbred and hybrid haploid inducers and heterosis regarding HIR, seed sets in cross-pollinations, plant and ear height, tassel size, and tassel branching across three years of haploid induction and (ii) assess hybrid inducers suitable for isolation fields. The information obtained in this study can help to determine whether and to what extent hybrid inducers are efficient for haploid induction in isolation fields. 

## 2. Results

### 2.1. Analysis of Variance

Inducer effects were highly significant for all observed traits. The interaction between year and inducer effects was significant for HIR, seed set, plant height, ear height, and tassel size, but it was not significant for the tassel branch. The year effect was not significant for all observed traits (Table 1).

### 2.2. Heterosis and Hybrid Performance

Mid-parent heterosis (MPH) was present for most hybrid inducers for HIR and seed sets over years (Table 2). In 2016, five of six crosses had a positive MPH ranging from 2.7% to 24.2%. In 2017, five of seven crosses had a positive MPH ranging from 1.5% to 65.7%. In 2018, however, the MPH was negative for most crosses ranging from −4.3% to −68.5%. 

Hybrid inducer BHI201/LH82-Ped128 had the highest positive MPH values in 2017 (65.7%) and 2018-1 (66.8%) for HIR. Another hybrid, BHI201/LH82-Ped126, had the highest positive MPH value in 2018-2 (65.8%). This genotype also had the second-highest positive MPH values in 2016 (23.1%) and 2017 (34.4%). On the other hand, hybrid inducer BHI201/Mo17-Ped115 had the highest negative MPH values in 2016 (−4.9%), 2018-1 (−68.5%), and 2018-2 (−45.4%) for HIR. This hybrid inducer also had the second-highest negative MPH value in 2017 (−10.5%).

The MPH on seed set was divergent, ranging from −47.3% to 145.5% over years. In 2017, four of seven crosses had negative MPH ranging from −47.3% to −5.1%. In 2018-1, however, the MPH was positive for all crosses ranging from 43.6 % to 145.5%. Regarding the seed set, the BHI201/LH82-Ped128 hybrid inducer showed the highest positive MPH in 2018-2 (49.2%), the second-highest positive MPH in 2016 (11.2%), the fifth-highest positive MPH value in 2018-1, but the highest negative MPH in 2017 (−47.3%). The BHI201/LH82-Ped126 hybrid inducer showed a positive MPH in 2016, 2018-1, and 2018-2. It had the highest positive MPH in 2016 (31.2%), whereas its negative MPH was only noticed in 2017 (6.1%).

All hybrid inducers resulted in taller plants than the inbred inducers as indicated by positive MPH on plant height ranging from 17.2% to 35.6% in 2017, from 25.3% to 45.8% in 2018-1, and from 21.2% to 36.9% in 2018-2 (Table 3). Regarding plant height, BHI201/LH82-Ped128 hybrid inducer had the highest positive MPH in 2017 (35.6 %) and 2018-2 (37.0%) and the second highest positive MPH in 2018-1 (41.9%). The BHI201/LH82-Ped126 hybrid inducer had the highest positive MPH in the 2018-1 period (45.8%), the third-highest positive MPH in 2018-2 (34.6%), and the lowest positive MPH in 2017 (17.2%) among the hybrid inducers observed.

All genotypes had higher ear positions due to the presence of positive MPH on ear height ranging from 2.6% to 56.4% in 2017, from 6.8% to 37.5% in 2018-1, and from 11.8% to 59.5% in 2018-2. Regarding ear height, the BHI201/LH82-Ped128 hybrid inducer showed the third-highest positive MPH in the 2018-1 period (27.8%) and the fourth-highest positive MPH in 2017 (23%) and 2018-2 (38.5%). The BHI201/LH82-Ped126 hybrid inducer showed the third-highest positive MPH in 2018-1 (22.2%) and 2018-2 (41.2%). The PHI-3/RWS hybrid inducer showed the highest positive MPH in 2017 (56.4%) and the lowest positive MPH in 2018-2 (11.8%).

Hybrid inducers showed bigger tassel size and tassel branching than their respective inbred parents (Table 4). In 2018, the MPH estimates of all crosses were positive, ranging from 7.7% to 32.3% for tassel size. Regarding tassel size, the BHI201/LH82-Ped128 hybrid inducer showed the third-highest positive MPH in 2018-1 (16.9%) and the fourth-highest positive MPH in 2018-2 (18.1%). The BHI201/LH82-Ped126 hybrid inducer showed the fourth-highest positive MPH in 2018-1 (9.6%) and the lowest positive MPH in 2018-2 (10.2%). The BHI201/LH82-Ped129 hybrid inducer showed the highest positive MPH in 2018-1 and 2018-2 (24.9% and 32.3%, respectively).

The MPH on (log) tassel branching was divergent, ranging from −5.3% to 13.1% in 2018. Regarding tassel branch, the BHI201/LH82-Ped126 hybrid inducer had the highest positive MPH in 2018-1 (13.1%) and negative MPH in 2018-2 (−0.4%). The BHI201/Mo17-Ped123 hybrid inducer showed the highest positive MPH in 2018-2 (13.0%) and the third-highest positive MPH in 2018-1 (10.9%). The PHI-3/RWS hybrid inducer showed the lowest negative MPH in 2018-1 and 2018-2 (−1.0 %, −5.3%, respectively).

Two of the seven crosses, BHI201/LH82-Ped126 and BHI201/LH82-Ped128, were promising hybrid inducers for haploid induction in isolation fields because they consistently had high HIR (7.1% to 30.5%) and moderate seed set (118 to 236), plant height (139.7 cm to 187.7 cm), and ear height (46.3 cm to 84.0 cm).

### 2.3. Inbred-Hybrid Relationship

The R-square value between the mid-parent (MP) and hybrid means was not significant for HIR (0.172) (Figure 1). In contrast, the value was significant for plant height (0.812), tassel size (0.581), and tassel branching (0.611). The result indicated that the performance of hybrid inducers could be predicted based on the MP values of correspondent parents for plant height, tassel size, and tassel branching. However, the MP-based hybrid prediction was not doable for HIR. 

### 2.4. Trait Correlations

Significant and positive correlation coefficients were noticed between plant height, ear height, and tassel size at both planting dates of 2018 (Table 5 and Table 6), ranging from 0.61 to 0.94, indicating that the taller plant stature led to possessing a higher ear position and larger tassel size in our haploid inducer genotypes. Haploid induction rate (HIR) was not correlated with the rest of the phenotypic traits, indicating that there was no trade-off between obtaining high HIR and good agronomic traits when breeding haploid inducers.

## 3. Discussion

Our study revealed that hybrid inducers had better vigor than inbred inducers over years regarding taller plants, higher ear position, and tassel size. However, for HIR, this superiority was not found for hybrid inducers except for the hybrids BHI201/LH82-Ped126 and BHI201/LH82-Ped128. We noticed that mid-parent heterosis was positive for all crosses for plant height, ear height, and tassel size. Negative heterosis was prevalent in some crosses for HIR, seed set, and tassel branching. Multiple studies have reported evidence of heterosis for agronomic traits of different corn types. Average mid-parent heterosis for plant height, ear height, and seed set was positive in sweet corn [14,15,16], waxy corn [17,18], and field corn [19,20]. In addition, the estimates of heterosis on tassel size and tassel branching were positive in field corn [21,22].

Heterosis reflects the restored vigor and productivity in crosses produced after a certain level of inbreeding. The dominance hypothesis was proposed as one option to explain the phenomenon of heterosis [23], where dominant genes have an overriding effect on dominant alleles. As hybrids contain a greater number of favorable dominant genes, they would be more vigorous than either of the parents with a smaller number of dominant alleles [24]. As the magnitudes of heterosis depend upon the parental genetic distance and the number of loci assembled in a hybrid [23], it was reasonable to obtain obvious levels of heterosis in each cross for plant stature and tassel size. Plant height and tassel size are polygenic traits, which are controlled by many known genes [25,26]. Likewise, HIR is a polygenic trait controlled by two major QTLs (*qhir1* and *qhir8*) and several minor QTLs [27]. So far, two major genes have been cloned, namely, *mtl* [5] and *zmdmp* [6]. 

We also noticed that there was a negative association between HIR and seed set over the trial years regarding the heterosis values. For instance, in trial years 2016 and 2017, the values of each hybrid were mostly positive for HIR whereas the values of the respective hybrids were mostly negative for seed set. In trial year 2018, the value of each hybrid was mostly negative for HIR whereas they were mostly positive for seed set. No significant differences in seed set between hybrid and inbred inducers were seen, probably because pollinations were conducted by hand, in which we ensured that enough pollen was being placed on donor silks to provide good pollination. If hybrid and inbred inducers were compared on isolations fields, hybrid inducers might produce better seed sets because they are taller and shed more pollen. The low seed set in this study was defined as the viable seeds of donor ears and calculated by adding the number of haploid and diploid seeds, excluding the number of either deformed or aborted seeds. The haploid induction rate was associated with embryo and endosperm abortion [28], although the correlation between those was not strongly positive [12]. The presence of deformed and aborted seeds would reduce the seed set after pollination. 

Breeders have the choice between lines, hybrids, and synthetics to develop new cultivars. Most of the haploid inducers available are inbred lines [9] which are associated with inbreeding depression. The reduced vigor and smaller morphological properties hinder inbred inducers from good adaptation to the new target environments and high efficiency under isolation fields. Currently, few hybrid inducers are released and available for licensing, for instance, RWS/RWK-76 [10], TAILs [11], and 2GTAILs [12]. Our two hybrid inducers, BH201/LH82-Ped126 and BH201/LH82-Ped128, were promising for haploid induction under isolation fields as they had moderate seed set, high HIR, plant stature, and ear height, moderate tassel size, and low tassel branching. The possible mechanism of haploid induction via wind pollination under isolation fields is that first hybrid inducers benefit heterosis for plant height, ear position, and tassel size to perform with better vigor and produce more pollen. As inducers act as male pollinators, more pollen is released by a single inducer plant, and higher rates of pollination success and seed set will be achieved. Moreover, taller inducer plants will lengthen the pollen dispersal, enabling breeders to reduce the male/female plant ratio and raise the population density of female donors. If so, more induced donor ears can be harvested, and higher haploid yields can be obtained.

As plant breeding is a numbers game [29], a higher number of genotypes evaluated will increase the possibility to obtain a favorable one. Therefore, further studies should include more possible combinations including the reciprocals to improve our hybrid ideotypes for the purpose of isolation fields because all hybrid inducers evaluated in this study had fewer phenotypic variabilities for plant height and tassel size than inbred inducers. It might be due to the maternal effect as those hybrid inducers shared the same female, which was BHI201, except for PHI-3/RWS. Previous studies reported the importance of reciprocal effects on agronomic traits [30,31] and seed set [32], as they correlated to attainable heterosis [33].

## 4. Materials and Methods

### 4.1. Plant Materials

Seven hybrid inducers and their respective inbred parents were evaluated for HIR and other traits of importance to maize maternal haploid inducers during the summers of 2016, 2017, and 2018. Nine inbred inducers were tested: BHI 201, three inducers with Mo17 background (Mo17-Ped115, Mo17-Ped117, and Mo-Ped123), three inducers with LH82 background (LH82-Ped126, LH82-Ped128, and LH82-Ped129), RWS, and PHI-3. The seven hybrid inducers tested were: (BHI201/Mo17-Ped115, BHI201/Mo17-Ped117, BHI201/Mo-Ped123, BHI201/LH82-Ped126, BHI201/LH82-Ped128, BHI201/LH82-Ped129, and PHI-3/RWS). The commercial hybrid Viking 60-01N, from Albert Lea Seed Company, was used as the only donor to evaluate HIR due to its good inducibility and ability to express the ‘*R1-nj*’ phenotype.

### 4.2. Experimental Design

The field trial was conducted under rainfed conditions at the Iowa State University Agronomy and Agricultural Engineering Farm, located in Boone (Iowa). The recommended practices for maize production in Central Iowa were followed. Pre- and post-emergent herbicides along with hoeing were used for weeding. Urea ammonium nitrate was applied in the area before sowing.

The inducers were not randomized because the great difference in vigor among them would adversely affect other traits for which data were collected for a companion study. That was also the reason for sowing closely related inducers side-by-side within each subblock. Inducer and donor blocks were sown side-by-side, and pollen from inducers in each subblock was carried to the adjacent donor subblock. Multicolored tags with easy-to-match codes were used to ensure that the pollen from each inducer plot was placed in the corresponding donor plot.

In the years of 2016 and 2017, the donor was planted at two different planting dates to ensure nicking with the inducers, which were planted at a single planting date. At each planting date of the donor, 18 seeds were sown in single-row plots. Inducers were sown in four row-plots of similar size and plant density. In 2018, the inducers and the donor were planted on two different planting dates (21 May and 31 May). The first planting date of the donor was exclusively pollinated by the first planting date of the inducer, whereas the second planting date of the donor was exclusively pollinated by the second planting date of the inducer. In 2018, 25 seeds of the inducers and donors were sown in 5.5 m long, single-row plots. The spacing between the rows was 0.75 m in all years. In all years, bulk pollen from each inducer plot was collected in tassel bags and used to pollinate at least 10 ears of the donor, which were covered before silk emergence using wax bags. Ears were harvested when the seeds reached the black layer stage and were air-dried for one week.

### 4.3. Data Collection

At the reproductive stage, the following agronomic traits were observed using the sample basis of three plants per plot: plant height (cm), as the distance from ground level to the node bearing the flag leaf; ear height (cm), as the distance from ground level to the node bearing the uppermost ear; tassel branching, as the total number of tassel branches including the primary and secondary branches; and tassel size (cm), as the distance from the lowermost primary branch to the tip of the tassel.

The visual haploid selection was performed using the *R1-nj* marker [34]. The seeds were classified as putative diploid if the seeds showed purple coloration on the endosperm and embryo and as putative haploid if the seeds showed purple endosperm and colorless embryo. The haploid induction rate (HIR) was calculated, as follows:(1)$$\mathrm{HIR}\;(\%)=\frac{\mathrm{seed}\;\mathrm{number}\;\mathrm{of}\;\mathrm{putative}\;\mathrm{haploid}}{\mathrm{total}\;\mathrm{seed}\;\mathrm{number}\;\mathrm{per}\;\mathrm{ear}}\;\times \;100$$

### 4.4. Statistical Analysis

Analysis of variance (ANOVA) was performed to assess if genotypes and years had a significant effect on HIR, seed set, plant height, and ear height, following the additive model, as follows:(2)$$Y_{ijk}=\mu +G_{i}+R_{j}\left(E_{k}\right)+E_{k}+GE_{ik}+\epsilon_{ijk}$$
where *Y_ijk_* is the phenotype of the *i*th inducer in *k*th year and *j*th replication, 𝜇 is the grand mean, *G*_𝑖_ is the genotypic effect of the *i*th inducer, *R_j_*(*E_k_*)** is the (random) replication effect of the *j*th replication nested within the *k*th year, *E_k_* is the environmental effect of the *k*th year, *GE_ik_* is the interaction effect of *i*th inducer and *k*th year, and *𝜀_𝑖𝑗𝑘_* is the residual error. 

The mid-parent heterosis (MPH) was calculated using the means of parental lines and hybrids following Rai [24]: MPH = (F_1_ − MP)/MP × 100% where F_1_ is the mean of the hybrid and MP is the averaged mean of the two inbred parents. Pearson linear correlation coefficients and linear regression coefficients [35] were calculated.

## 5. Conclusions

The haploid induction rate (HIR), seed set, plant height, ear height, tassel size, and tassel branching of haploid inducer were significantly influenced by genotype, year, and the interaction between genotype and year. The distinct differences between the two inducer types were noticed in plant height, ear height, and tassel size where hybrid inducers benefit heterosis values over their inbred parents. We suggested two hybrid inducers, BH201/LH82-Ped126 and BH201/LH82-Ped128, to be adopted for haploid induction in isolation fields as they had moderate seed set, high HIR, plant stature, and ear height, and moderate tassel size. Although substantial heterosis was not prevalent, neither hybrid means which could not significantly surpass inbred means on HIR, the concept of hybrid breeding was still helpful for breeding haploid inducers as it offers convenience and resource effectiveness for haploid induction by means of improving plant vigor without compromising the haploid frequency.

## Figures and Tables

**Figure 1 plants-12-01095-f001:**
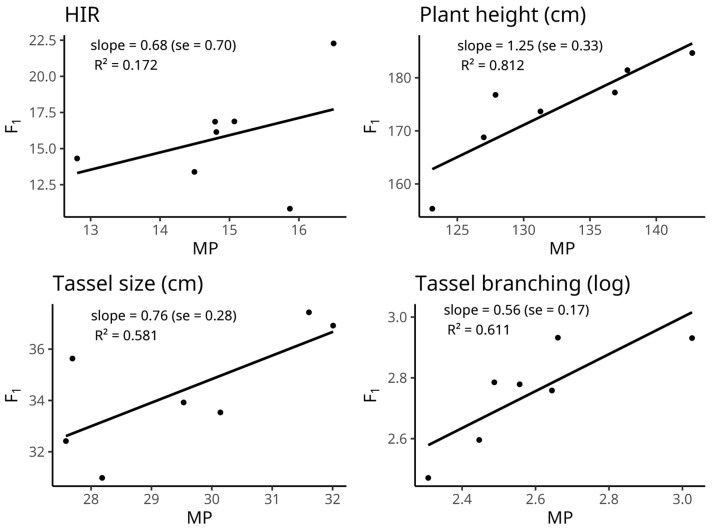
Linear regressions of hybrid performance (F_1_) on mid-parent (MP), for haploid induction rate, plant height, tassel size, and tassel branching. Estimates (and s.e.’s) of the slopes averaged over the years, and R^2^ values are shown. F_1_ and MP values are also averaged over the years.

**Table 1 plants-12-01095-t001:** Analysis of variance across years between 2016 and 2018. F-ratios are shown for each source along with the numerator and Kenward-Roger adjusted denominator degrees of freedoms (in parentheses).

Source	HIR	Seed Sets	Plant Height	Ear Height	Tassel Size	(Log) Tassel Branch
Year	39.7 (3, 8.1)	135.0 (3, 8.1)	111.9 (2, 6)	141.89 (2, 6)	14.48 (1, 4)	0.03 (1, 4)
Inducer	22.4 *** (15,117)	16.13 *** (15, 117)	338.0 *** (15, 90)	38.84 *** (15, 90)	38.97 *** (15, 59)	15.47 *** (15, 59)
Year × Inducer	11.0 *** (44, 117)	6.58 *** (44, 117)	14.6 *** (30, 90)	4.25 *** (30, 90)	1.94 * (15, 59)	1.19 ^ns^ (15, 59)
**Variance components estimates:**						
Rep (Year)	0.0466	0	2.43	0	0.39	0
Error	4.832	668	19.70	34.7	3.42	0.0222

* *p*-value < 0.05; *** *p*-value < 0.001; ^ns^
*p*-value > 0.1.

**Table 2 plants-12-01095-t002:** Mid-parent (MP), hybrid mean (F1), and mid-parent heterosis (MPH) of hybrid haploid inducers on haploid induction rate (%) and seed set between 2016 and 2018.

Year	Hybrid Inducers	Haploid Induction Rate (%)			Seed Set (Seeds Per Donor Ear)		
		MP	F_1_	MPH (%)	MP	F_1_	MPH (%)
16	BHI201/LH82-Ped126	12.1	14.9	23.1	179.8	236.0	31.2
	BHI201/LH82-Ped128	15.9	16.3	2.7	190.3	211.7	11.2
	BHI201/LH82-Ped129	13.9	16.4	18.2	176.2	169.3	−3.9
	BHI201/Mo17-Ped115	15.3	14.5	−4.9	255.3	226.7	−11.2
	BHI201/Mo17-Ped117	13.8	17.1	24.2	208.7	185.7	−11
	BHI201/Mo17-Ped123	11.8	13.6	15.1	151.0	167.3	10.8
17	BHI201/LH82-Ped126	16.9	22.7	34.4	225.3	211.7	−6.1
	BHI201/LH82-Ped128	15.0	24.8	65.7	227.2	119.7	−47.3
	BHI201/LH82-Ped129	15.7	12.8	−18.3	227.2	215.7	−5.1
	BHI201/Mo17-Ped115	17.8	16.0	−10.5	205.2	235.3	14.7
	BHI201/Mo17-Ped117	19.7	20.8	5.4	205.0	206.7	0.8
	BHI201/Mo17-Ped123	18.1	21.1	16.9	216.3	201.3	−6.9
	PHI-3/RWS	14.1	14.3	1.5	169.2	244.7	44.6
18-1	BHI201/LH82-Ped126	17.5	7.1	−59.5	61.2	118.0	92.9
	BHI201/LH82-Ped128	18.3	30.5	66.8	85.8	165.3	92.6
	BHI201/LH82-Ped129	15.4	12.4	−19.2	66.8	131.3	96.5
	BHI201/Mo17-Ped115	16.1	5.1	−68.5	84.3	150.3	78.3
	BHI201/Mo17-Ped117	13.8	15.2	10.5	91.0	130.7	43.6
	BHI201/Mo17-Ped123	15.4	20.9	35.9	75.0	155.7	107.6
	PHI-3/RWS	14.5	13.1	−9.7	119.3	293.0	145.5
18-2	BHI201/LH82-Ped126	13.7	22.8	65.8	97.5	134.0	37.4
	BHI201/LH82-Ped128	16.9	17.5	3.7	103.0	153.7	49.2
	BHI201/LH82-Ped129	13.1	12.0	−8.7	83.8	111.0	32.4
	BHI201/Mo17-Ped115	14.3	7.8	−45.4	133.5	109.7	−17.9
	BHI201/Mo17-Ped117	12.0	11.5	−4.3	112.8	162.7	44.2
	BHI201/Mo17-Ped123	13.9	11.8	−14.9	97.5	132.3	35.7
	PHI-3/RWS	9.8	15.5	58.8	119.8	149.3	24.6

**Table 3 plants-12-01095-t003:** Mid-parent (MP), hybrid mean (F1), and mid-parent heterosis (MPH) of hybrid haploid inducers on plant height (cm) and ear height (cm) between 2017 and 2018.

Year	Hybrid Inducers	Plant Height (cm)			Ear Height (cm)		
		MP	F_1_	MPH (%)	MP	F_1_	MPH (%)
17	BHI201/LH82-Ped126	119.1	139.7	17.2	44.2	46.3	4.8
	BHI201/LH82-Ped128	115.8	157.0	35.6	48.2	59.3	23
	BHI201/LH82-Ped129	115.0	147.0	27.9	44.4	57.7	29.9
	BHI201/Mo17-Ped115	127.1	166.0	30.6	46.1	59.7	29.6
	BHI201/Mo17-Ped117	130.8	159.3	21.8	49.1	50.3	2.6
	BHI201/Mo17-Ped123	127.1	165.7	30.3	49.9	68.3	37.0
	PHI-3/RWS	118.2	156.7	32.6	36.7	57.3	56.4
18-1	BHI201/LH82-Ped126	127.3	185.7	45.8	42.8	52.3	22.2
	BHI201/LH82-Ped128	130.8	185.7	41.9	42.0	53.7	27.8
	BHI201/LH82-Ped129	141.0	189.3	34.3	44.0	56.3	28.0
	BHI201/Mo17-Ped115	141.7	186.0	31.3	42.7	58.7	37.5
	BHI201/Mo17-Ped117	151.8	199.3	31.3	51.3	61.0	18.8
	BHI201/Mo17-Ped123	151.2	193.7	28.1	56.2	60.0	6.8
	PHI-3/RWS	120.0	150.3	25.3	34.7	39.3	13.5
18-2	BHI201/LH82-Ped126	134.5	181.0	34.6	57.8	81.7	41.2
	BHI201/LH82-Ped128	137.0	187.7	3.0	60.7	84.0	38.5
	BHI201/LH82-Ped129	137.8	184.7	34.0	55.8	82.7	48.1
	BHI201/Mo17-Ped115	141.8	179.7	26.7	60.8	80.3	32.1
	BHI201/Mo17-Ped117	145.5	195.3	34.2	59.0	77.0	30.5
	BHI201/Mo17-Ped123	135.2	185.0	36.9	58.5	93.3	59.5
	PHI-3/RWS	131.2	159.0	21.2	53.7	60.0	11.8

**Table 4 plants-12-01095-t004:** Mid-parent (MP), hybrid mean (F1), and mid-parent heterosis (MPH) of hybrid haploid inducers on tassel size (cm) and tassel branching in 2018.

Year	Hybrid Inducers	Tassel Size (cm)			(log) Tassel Branching		
		MP	F_1_	MPH (%)	MP	F_1_	MPH (%)
18-1	BHI201/LH82-Ped126	27.7	30.3	9.6	2.3	2.7	13.1
	BHI201/LH82-Ped128	26.7	31.2	16.9	2.2	2.4	8.0
	BHI201/LH82-Ped129	27.1	33.8	24.9	2.6	2.9	10.8
	BHI201/Mo17-Ped115	30.5	32.8	7.7	2.5	2.8	10.3
	BHI201/Mo17-Ped117	31.5	35.3	12.2	2.6	2.9	11.3
	BHI201/Mo17-Ped123	30.4	32.8	7.9	2.5	2.7	10.9
	PHI-3/RWS	29.6	35.8	21.3	3.1	3.1	−1.0
18-2	BHI201/LH82-Ped126	28.7	31.6	10.2	2.5	2.5	−0.4
	BHI201/LH82-Ped128	28.5	33.7	18.1	2.4	2.5	6.2
	BHI201/LH82-Ped129	28.3	37.4	32.3	2.7	2.7	−1.9
	BHI201/Mo17-Ped115	29.8	34.2	14.9	2.6	2.8	7.2
	BHI201/Mo17-Ped117	32.5	38.5	18.4	2.7	3.0	9.1
	BHI201/Mo17-Ped123	28.7	35.0	22.2	2.5	2.8	13.0
	PHI-3/RWS	33.7	39.0	15.9	3.0	2.8	−5.3

**Table 5 plants-12-01095-t005:** Linear correlation among phenotypic traits of 16 haploid inducer genotypes evaluated at the first planting date of 2018.

Traits	Seed Set	HIR	Plant Height	Ear Height	Tassel Size
HIR	0.09				
Plant height	0.10	0.10			
Ear height	−0.11	0.07	0.90 **		
Tassel size	0.36	0.06	0.70 **	0.70 **	
Tassel branching	0.40	−0.32	0.02	−0.10	0.33

** r is significant at *p* ≤ 0.01. Darker colors indicate higher correlation coefficients.

**Table 6 plants-12-01095-t006:** Linear correlation among phenotypic traits of 16 haploid inducer genotypes evaluated at the second planting date of 2018.

	Seed Set	HIR	Plant Height	Ear Height	Tassel Size
HIR	−0.17				
Plant height	0.07	0.38			
Ear height	−0.06	0.44	0.94 **		
Tassel size	−0.05	0.03	0.61 **	0.43	
Tassel branching	0.15	−0.63	0.03	−0.14	0.38

** r is significant at *p* ≤ 0.01. Darker colors indicate higher correlation coefficients.

## Data Availability

Data are available within this article.

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
