# Peer review of "A Comparison between Inbred and Hybrid Maize Haploid Inducers"

_plants, 2023, doi:10.3390/plants12051095_

Round 1

Reviewer 1 Report

The authors studied how seed set, plant height, ear height, tassel size, and tassel branch affect the production of maize haploid seeds, and also studies the correlation between these traits. The manuscript is suitable to be published on Plants. However, the manuscript is not easy to understand for readers who do not work in this area. Below are some suggestions to make it easy to understand
1. Add details about the possible mechanisms the traits affect haploid seed production
2. Add some details on how to distinguish haploid seeds and diploid seeds
3. It took me a few minutes to understand the sentence "Induction of haploidy requires genotypes with inherent ability to stimulate haploid embryos (inducers) used as males and genotypes for which haploids are necessary (donors) as females"

Author Response

The authors studied how seed set, plant height, ear height, tassel size, and tassel branch affect the production of maize haploid seeds, and also studies the correlation between these traits. The manuscript is suitable to be published on Plants. However, the manuscript is not easy to understand for readers who do not work in this area. Below are some suggestions to make it easy to understand:

Point 1: Add details about the possible mechanisms the traits affect haploid seed production.

Response 1: We thank the reviewer for pointing out this issue. Revision has been made by adding several sentences explaining the possible mechanisms of traits observed affecting the haploid production. We have mentioned how heterosis advantage in hybrids can improve the efficiency of haploid production under isolation fields. The revision can be seen in page 9 lines 250-258.

Page 9 lines 250-258:

The possible mechanism of haploid induction via wind-pollination under isolation fields is that first hybrid inducers benefit heterosis for plant height, ear position, and tassel size to perform better vigor and produce more pollens. Since inducer acts as male pollinator, the more pollens are released by a single inducer plant, the higher rates of pollination success and seed set will be achieved. Besides, taller inducer plant will lengthen the pollen dispersal, enabling breeders to reduce the male/female plant ratio and to raise the population density of female donors. If so, more induced donor ears can be harvested, and higher haploid yields can be obtained.

Point 2: Add some details on how to distinguish haploid seeds and diploid seeds.

Response 2: We thank the reviewer for pointing out this issue. The haploid and diploid seeds can be distinguished via visual selection regarding anthocyanin marker at seed stage. Haploid seeds were indicated by the seeds with purple endosperm and colorless embryo, while diploid seeds were indicated by the seeds with purple endosperm and embryo (Nanda and Chase, 1966). We have previously mentioned the brief description on how the expression of R1-nj marker was determined in the materials and methods part “Data Collection” (page 10 lines 313-317). We hope that these explanations are clearer.

Point 3: It took me a few minutes to understand the sentence "Induction of haploidy requires genotypes with inherent ability to stimulate haploid embryos (inducers) used as males and genotypes for which haploids are necessary (donors) as females".

Response 3: We thank the reviewer for pointing out this issue. Revision has been made by modifying that sentence to make it clearer.

Page1 lines 28-30:

Through in vivo maternal system, haploid induction requires haploid inducers, a male genotype having ability to induce haploids, and source germplasm as donor genotypes.

Reviewer 2 Report

In the introduction line 46, what does R1-nJ represent marker, or what? Please make it more clear.

The results are precise, but after checking the data in the Table, there are some considerable differences in data as, in the seed set in 2016, the value represents 255 while the low one is 151. Can you make sure?

The results line (23.1 %) has a space between the digits and the percentage. Please remove or add the space to all to keep the same format. Such as, in line 92, there is no space between the digits and percentage.   

Similarly, follow the same pattern for the 136 lines and all manuscripts.

Figure the quality of the Figure needs to be clarified. Please improve it.

Please check line 212, Negative ……..remove the space before the words. 

Please check the journal format for table.

Please confirm all references cite in the text present in references section.

Author Response

Point 1: In the introduction line 46, what does R1-nJ represent marker, or what? Please make it more clear.

Response 1: We thank the reviewer for pointing out this issue. The R1-nj represents phenotypic marker, commonly used in maize for identifying haploid/diploid at kernel stage. The R1-nj marker is an anthocyanin color marker particularly expressed on both the crown endosperm and the scutellum embryo of kernels (Nanda and Chase, 1966). Revision has been made by modifying the sentence previously appeared in line 46, as follows:

Page 1 line 45:

However, if discrimination is based on R1-nj anthocyanin kernel marker, all inducer types carrying this single gene are equally suitable.

Point 2: The results are precise, but after checking the data in the Table, there are some considerable differences in data as, in the seed set in 2016, the value represents 255 while the low one is 151. Can you make sure?

Response 2: We thank the reviewer for pointing out this issue. We have made sure that the data all are correct. Since MP values derived from the average of two pairwise parents of a respective hybrid, the considerable differences in MP values among hybrids in 2016, for instance, between 255.3 in hybrid BHI201/Mo17-Ped115 and 151.0 in hybrid BHI201/Mo17-Ped123 are due to genotypic effects. Both hybrid inducers shared the same female parent (BHI201), but they did have the different male parents (Mo17-Ped115 vs Mo17-Ped123); thus, the MP values of those two hybrids will be considerably different. We hope that these explanations are clearer.

Point 3: The results line (23.1 %) has a space between the digits and the percentage. Please remove or add the space to all to keep the same format. Such as, in line 92, there is no space between the digits and percentage. Similarly, follow the same pattern for the 136 lines and all manuscripts.  

Response 3: We thank the reviewer for pointing out this issue. We have rechecked carefully and removed the space between the digits and the percentage wherever they appear throughout the manuscript to keep the same format. The revision can be seen in the revised manuscript and highlighted in blue. 

Point 4: Figure the quality of the Figure needs to be clarified. Please improve it.

Response 4: We thank the reviewer for pointing out this issue. The image quality has been improved. The current image is a 6”x8” 600dpi PNG image provided in tiff format that can be seen in the revised manuscript (page 7 lines 168-172).

Point 5: Please check line 212, Negative ……..remove the space before the words.

Response 5: We thank the reviewer for pointing out this issue. The space before the words has been removed, and the revision can be seen in page 8 line 210.

Point 6: Please check the journal format for table.

Response 6: We thank the reviewer for pointing out this issue. We have checked the journal format for table. Revision has been made for Table 1 (page 2 lines 73-75), Table 2 (page 3 lines 84-86), Table 3 (page 5 lines 139-141), and Table 4 (page 6 lines 152-154).

Point 7: Please confirm all references cite in the text present in references section.

Response 7: We thank the reviewer for pointing out this issue. We have confirmed that all references cited in the text are present in references.